# Completeness and validity of alcohol recording in general practice within the UK: a cross-sectional study

Kathryn Mansfield ,[1] Elizabeth Crellin,[2] Rachel Denholm,[3] Jennifer K Quint ,[4] Liam Smeeth,[1] Sarah Cook,[1] Emily Herrett[1]

KM and EC are joint first authors.

[1]Department of Non-Communicable Disease Epidemiology, London School of Hygiene and Tropical Medicine, London, UK
[2]Clinical Practice Research Datalink, London, UK
[3]Centre for Academic Primary Care, University of Bristol, Bristol, UK
[4]Respiratory Epidemiology,Occupational Medicine and Public Health, National Heart and Lung Institute, Imperial College, London, UK

**Correspondence to**
Dr Emily Herrett;
emily.herrett@lshtm.ac.uk

## ABSTRACT

**Background** Since 2010 the National Institute for Health and Care Excellence has recommended screening adults for excessive alcohol consumption to try and help prevent alcohol-use disorders. Little is known about the extent to which these recommendations are followed, and the resulting completeness and validity of alcohol-related data recording in primary care.

**Objective** To investigate the completeness and accuracy of recording of alcohol use within primary care records in the UK.

**Design and setting** Cross-sectional study in the Clinical Practice Research Datalink.

**Participants** We included all adult patients registered on 1st January 2018 with ≥1 year of follow-up.

**Primary and secondary outcome measures** We calculated prevalence of alcohol consumption recording overall and within patient groups. We then validated alcohol consumption data against recommended screening tools (Alcohol Use Disorders Identification Test (AUDIT)) as the gold standard. We also calculated how prevalence of alcohol recording changed over the preceding decade.

**Results** In 2018, among 1.8 million registered adult patients, just over half (51.9%) had a record for a code related to alcohol in the previous 5 years. Recording of alcohol consumption was more common among women, older people, ex-smokers and those from more deprived areas, who were overweight/obese, or with comorbidities. A quarter of patients had units per week recorded in the last 5 years, but <10% had an AUDIT or Fast Alcohol Screening Test (FAST) alcohol screening test score. The recorded alcohol measures corresponded to results from gold standard AUDIT scores. The distribution of consumption among current drinkers was similar to the Health Survey for England.

**Conclusions** Half of adults in UK primary care have no recorded alcohol consumption data. When consumption is recorded, we have demonstrated internal and external validity of the data, suggesting greater recording may help identify opportunities for interventions to reduce harms.

## INTRODUCTION

Hazardous alcohol use is a key behavioural risk factor affecting health. It is estimated that over 20% of adults in England are drinking at levels that are considered harmful,[1] with a resulting cost to the National Health Service

### Strengths and limitations of this study

► This study investigated alcohol recording across nearly 1.8 million UK adults in primary care, providing the power to understand in detail how alcohol use is recorded in general practice.
► We demonstrated the validity of alcohol recording internally within general practice records, and externally compared with the Health Survey for England.
► Though we demonstrated levels of alcohol recording and screening, it does not mean that general practitioners have acted on the screening result, or that they have screened at an appropriate time.

(NHS) of over £3 billion per year.[2] In the age group 15 to 49 years, alcohol is the leading risk factor for ill health, early death and disability.[3]

Identification of high-risk patients can usefully be undertaken in primary care, where appropriate interventions can be promoted. The Alcohol Use Disorders Identification Test (AUDIT)[4] is the main tool for alcohol screening recommended by National Institute for Health and Care Excellence (NICE), though shortened versions (such as AUDIT-C and Fast Alcohol Screening Test (FAST)) are also considered acceptable.[5] NICE guidelines published in 2010 recommend that opportunistic screening of the general population should take place during new patient registrations, and when screening or managing other conditions, promoting sexual health, during antenatal appointments and treating minor injuries. In addition, guidelines recommend screening 'high risk' groups, including those with relevant physical conditions (such as hypertension) and relevant mental health problems (such as depression).[5] Additionally, the NHS health check, which is offered to all people in the UK aged 40 to 74 years every 5 years, includes the AUDIT questions.[6]

Khadjesari *et al*[7] showed relatively high levels of alcohol recording (76%) among

new registrants in primary care between 2007 and 2009, although the use of validated screening tools was low (9%). They did not report levels for the whole registered patient population, which includes those who are less mobile and not subject to new patient registrations, nor specifically among patients with conditions that may be associated with, or exacerbated by, high-risk drinking. Poor recording of alcohol use has two major impacts: first and most importantly, patients may not be receiving appropriate interventions; and second, for researchers, poor recording may prohibit reliable studies of the impact of alcohol on health using these data. Indeed, some recent studies using alcohol data in a large UK-based primary care electronic health record database (Clinical Practice Research Datalink (CPRD)) have indicated that bias may have arisen due to poor recording of alcohol data.[8 9] For researchers using the data it is important to know not just whether screening occurs but if there are particular subgroups more likely to be screened and to what extent alcohol codes used are meaningful reflections of actual alcohol use.

Using primary care data from the CPRD, this study assessed the utility of recorded alcohol data in primary care by assessing completeness of recording of alcohol status at different levels and by various patient level factors, and comparing measures of alcohol recorded in the CPRD internally and to external sources. Our study also offers insight into the completeness of alcohol data in primary care since the introduction of 2010 NICE guidance,[5] including specifically among those identified as being at higher risk according to NICE recommendations, and provides guidance for future studies related to alcohol use using primary care data.

## METHODS

### Study design and setting

We undertook a cross-sectional study with data from the Clinical Practice Research Datalink GOLD database, an ongoing primary care database containing the anonymised medical records from general practitioners in the UK using the Vision software to capture health record data with coverage of 15.4 million patients. Data from CPRD includes diagnoses, tests, clinical measurements, prescriptions and specialist referrals. CPRD uses Read morbidity coding to summarise each patient encounter with a code (or codes) that correspond to a standard set of clinical terminology. Roughly half of all practices in the CPRD have consented to data linkage and therefore deprivation data (based on quintiles of the Indices of Multiple Deprivation) is available for these practices. The study included all patients in CPRD GOLD who were 18 years or over, alive and registered, for at least 1 year, with a CPRD GOLD practice on 1st January 2018.

### Factors associated with alcohol recording

We investigated a range of potential factors that may be associated with alcohol-use recording in primary care: age,

sex, deprivation, geographical region, selected comorbidities related to alcohol use that could be captured in electronic health record data (liver disease, hypertension, depression and anxiety) or those where health function monitoring is likely to be more frequent/complete (diabetes mellitus), time since registration at the current practice (1 to 5 years vs ≥5 years), body mass index (BMI), smoking status and ethnicity.

We identified liver disease, hypertension and diabetes mellitus based on presence of a recorded diagnosis at any time prior to the study date. We identified depression and anxiety based on the presence of a diagnosis at any time within the 5 years prior to the study date (1st January 2018). We used existing morbidity code lists and algorithms to define smoking status, BMI and ethnicity.[10 11]

### Outcomes

Our main outcomes were the prevalence of alcohol-use recording in the 5 years prior to the study date, 1st January 2018. Prevalence of alcohol recording included any record of alcohol use or its effects, and we also broke it down into the following categories:

i. Any code suggesting that alcohol was discussed in the consultation.
ii. Codes indicating AUDIT (including short-form AUDIT-C and full AUDIT) or FAST screening.
iii. Codes quantifying alcohol use, comprised of:
   a. drinking status (current drinker, ex, non; either as recorded in Read codes, or in structured data),
   b. level of drinking (non, light, moderate, heavy drinker; as recorded in Read codes),
   c. units per week,
   d. AUDIT, AUDIT-C or FAST scores (AUDIT categorised according to WHO guidelines on risk level: 0 to 7 – non-drinker or low-risk drinking; 8 to 15 – hazardous drinking; 16 to 19 – harmful drinking; 20 to 40 – possible dependence[12]).

We have made all morbidity code lists used in this study available to download (https://doi.org/10.17037/data.00001071). We have also provided further detail regarding variable definitions in the online supplementary appendix (online supplementary text 1 and online supplementary table 1).

We considered the record of a score from an alcohol-screening test using a validated tool (AUDIT, AUDIT-C or FAST) as the gold standard for alcohol recording. The AUDIT score is a 10-item instrument with a maximum score of 40 designed specifically for screening for hazardous or harmful alcohol use in primary care.[12 13] A score of 8 or more on the AUDIT tool is associated with harmful or hazardous drinking, although a slightly lower threshold can be used in women. The AUDIT tool can be abbreviated in time-limited settings to the first three questions (dealing with alcohol consumption only) (AUDIT-C) to give a score out of 12 or a four-item screening tool (FAST), which give a score out of 16. These abbreviated tools have lower thresholds to indicate harmful or hazardous drinking than the standard AUDIT tool:

AUDIT-C — three for women and four for men or greater than 5 for high risk drinking; and FAST — three.[14–16] Due to differences in scoring systems and different threshold scores to identify problem linking between the AUDIT, AUDIT-C and FAST tools, we restricted our internal validation to those with the full AUDIT or AUDIT-C score.

## Statistical analysis
### Completeness of alcohol recording
To investigate the completeness of alcohol recording, we estimated the proportion of patients with a record for one or more morbidity codes relating to alcohol in the categories outlined above (ie, (i) Any code suggesting that alcohol was discussed in the consultation; (ii) Codes indicating AUDIT, AUDIT-C or FAST screening or (iii) Codes quantifying alcohol use) in the 5 years prior to 1st January 2018 (each patient could have codes recorded in more than one category). We then described alcohol recording prevalence in strata of individual characteristics (ie, age, sex, and other factors described above under the heading 'Factors associated with alcohol recording'). We also described the frequency of consultations with a record of an alcohol code in the 5 years before the study date, and the number of days from a patient's registration to their first record of an alcohol use code.

### Internal validity
We validated codes quantifying the level of alcohol use (ie, current/ex/non; light/moderate/heavy; units/week (Category iii above)) against an AUDIT score from the *same date* and the *same patient*. Full AUDIT versus AUDIT-C screening were considered separately. We calculated median AUDIT and AUDIT-C scores for each category of level of alcohol use recording: (a) drinking status, (b) drinking level and (c) units per week.

Similarly, we also validated codes indicating drinking status, drinking level and AUDIT/AUDIT-C scores against the number of units per week. For each variable, we only included a patient's most recent record (to the study date).

### External validity
We identified current alcohol-use status for each individual with a relevant primary care alcohol record by identifying the most recent record indicating alcohol intake; that is, either a morbidity code indicating alcohol-use status (classified as current or non-/ex-drinkers) or a record indicating the number of units of alcohol consumed per week. We described current alcohol-use status separately for men and women from CPRD, and compared the results to the Health Survey for England 2016.[17] Then, for CPRD patients with a recorded 'current' drinking status, we grouped the recorded number of units consumed per week into the following categories: men ≤14 (lower risk), 15 to 50 (increased risk), >50 (higher risk); women ≤14 (lower risk), 15 to 35 (increased risk), >35 (higher risk), and again compared these data to the Health Survey for England data 2016.[17]

### Time trends in alcohol recording
To understand how alcohol recording has been changing over time, we carried out a cross-sectional study on the 1st January each year between 2009 and 2018. We estimated the number of patients who would be eligible for the study, using the same criteria as above (registered for 1 year at a CPRD practice, and aged 18 or over on the study date), and calculated the proportion of patients with a record of alcohol consumption (based on a record of drinking status), and level of drinking (based on Read codes, units per week, AUDIT and AUDIT-C scores) over the past 5 years.

### Sensitivity analyses
1. In the main analysis, we restricted the validation of alcohol codes to cases where patients had an AUDIT/AUDIT-C score recorded on the same date as another alcohol code of interest. In sensitivity analyses, we included cases where any AUDIT/AUDIT-C score was recorded up to 30 days before or after the code of interest.
2. In the main analysis, we recorded liver disease, hypertension and diabetes mellitus based on presence of a diagnosis at any time prior to the study date. We tested this definition by repeating the analysis limiting to diagnoses of these comorbidities within 5 years prior to the study date.
3. In the main analysis, we recorded depression and anxiety based on presence of a diagnosis within 5 years prior to the study date. We tested this definition by repeating the analysis limiting to diagnoses of these comorbidities within 12 months prior to the study date.

Data were analysed using Stata V.14 (StataCorp, Texas, USA).

## Patient involvement
Neither patients nor the public were involved in the design or analysis of the study. This work uses data provided by patients and collected by the UK National Health Service as part of their care and support.

## RESULTS
There were 1 768 651 adults aged 18 years or over, alive and registered, for at least 1 year, with a CPRD practice on 1st January 2018, and therefore included in the study; 49.4% were male, and their mean age was 49.7 years. The mean duration of registration prior to the study date was 18.7 years (SD 14.5 years). One-fifth had hypertension, 2.0% had liver disease, 6.8% had diabetes, 6.7% had depression and 4.6% had anxiety.

## Prevalence
Overall, 918 254 (51.9%) patients had a record indicating that alcohol was discussed within the last 5 years. This included diagnoses of alcohol use disorders, their management, harm due to alcohol consumption, in addition to the consumption of alcohol.

## Drinking status

Drinking status (current, ex-, non-drinker) was recorded in 862 330 (48.8%) of patients, though this was not consistent across patient groups. Recording of drinking status was higher in women compared with men (52.7% in women vs 44.7% in men), increased with age (30.6% in 18 to 24 years, 62.2% in 75+years) and was more common among recently registered patients (in the last 1 to 5 years) than in those registered for 5 or more years (73.6% vs 43.0%, respectively). There were also differences in recording between geographical region (higher recording in the North of England, lower recording in Northern Ireland and Wales), by deprivation (recording was more common in deprived areas) and by ethnicity (highest for white patients). Patients who had missing smoking or BMI data had particularly low levels of alcohol status recorded, but even in those with smoking status recorded, drinking status was only recorded in approximately 50% to 60% of patients. If smoking and BMI were recorded there were differences in the recording of alcohol status at different levels of smoking/BMI: for example, ex and current smokers were more likely to have alcohol status recorded than non-smokers, as were patients who were overweight. Patients were also more likely to have alcohol status recorded with each of the morbidities investigated (liver disease, hypertension, diabetes, depression and anxiety). Patients with diabetes were the most likely to have alcohol levels recorded (82.3% had an alcohol record) (table 1).

## Level of consumption

Overall, 862 642 (48.8%) of adult patients had some level of alcohol consumption (including non-drinkers) recorded in the previous 5 years, identified through recording of Read codes (heavy, moderate, light or non-drinker), units per week or formal alcohol screening score using AUDIT, AUDIT-C or FAST.

Level of alcohol consumption data were recorded using Read coding for 31.6% of patients, in units per week for 24.4% of patients and via a record of formal alcohol screening for 11.5% of patients with a Read code indicating that they had completed an AUDIT, AUDIT-C or FAST questionnaire, but only 8.5% of patients had the actual score available.

Recording of level of alcohol consumption by patient and practice characteristics are shown in table 1. Patients with diabetes had the highest levels of alcohol consumption recording, but only 39.5% of them had a recording of units per week, and only 10.1% had a recorded AUDIT, AUDIT-C or FAST score.

Among patients with a recorded drinking status, 20.3% were non-drinkers, 3.7% ex-drinkers and 76.1% current drinkers. table 2 describes levels of drinking among patients for whom it was recorded.

If we use all data (from all levels of alcohol consumption categories, that is, coded heavy/moderate/light/non-drinker, recorded units per week or formal alcohol screening score) to derive drinking status based on recorded status, units per week and AUDIT scores, an additional 35 105 patients (2.0% of eligible patients) have a drinking status. This derived status shows that 20.8% are non-drinkers, 3.5% ex-drinkers and 75.7% are current drinkers.

## Internal validity

In our study population, there were 77 212 (4.4%) patients with at least one AUDIT score record available, 21 099 (1.2%) patients with at least one AUDIT-C score record available and 431 394 (24.4%) with units-per-week records. table 2 describes the cross validation between alcohol measures and AUDIT scores, AUDIT-C scores and units per week.

## Drinking status

Current drinking (based on Read coding) was associated with a median AUDIT-C score of 3 (IQR 2 to 5), which is under the standard threshold defining risky drinking, and non-/ex-drinkers had a median AUDIT-C score of 0. For those with full AUDIT scores, the non- and ex-drinkers had median scores of 0, and current drinkers had a score of 3 (IQR 1 to 5). On average, current drinkers drank six units per week (IQR 2 to 14).

## Drinking level

Light and moderate drinkers had low AUDIT and AUDIT-C scores, while heavy drinkers had a median AUDIT-C score of 8 (IQR 6 to 9) and a median AUDIT score of 6 (IQR 5 to 9), which indicates alcohol education. Light drinkers drank three units per week (IQR 1 to 8), moderate drinkers six per week (IQR 2 to 14) and heavy drinkers 18 per week (IQR 9 to 30).

## Units per week

Units per week increased with increasing AUDIT and AUDIT-C scores, but even the highest category of units per week (43+) only had a median AUDIT score of 11 (IQR 8 to 12), which indicates simple alcohol advice.

## AUDIT scores

The majority (92.5%) of patients had scores in the lowest category (AUDIT score 0 to 7, where alcohol education is recommended according to WHO guidelines[12]). Patients with higher AUDIT scores consumed more units per week; for example, those in the lowest AUDIT score category (0 to 7, alcohol education) had a median units per week of four (IQR 2 to 10) compared with a median of 55 units per week (IQR 32 to 70) in the highest AUDIT score category (score 20 to 40, indicating need for referral to specialist for diagnostic evaluation and treatment according to WHO guidelines[12]).

## AUDIT-C scores

Patients with low risk AUDIT-C scores (<5) consumed fewer units per week; median four units per week (IQR 2 to 8) compared with a median of 12 units per week (IQR 6 to 20) in the high risk category (score ≥5). Where there were records for both AUDIT and AUDIT-C, in those with high risk AUDIT-C scores (which would indicate the need

**Table 1** Characteristics of adult patients registered in the CPRD, on 1st January 2018, and those with records indicating drinking status, level of consumption, units per week and AUDIT/FAST scores

| | Study population: All individuals 18+ years registered with a CPRD practice on 1st January 2018* | Individuals in the study population with recording of alcohol use in different categories in the 5 years before 1st January 2018† | | | | | | | |
| --- | --- | --- | --- | --- | --- | --- | --- | --- | --- |
| | | Has a record for any read code relating to alcohol use | Has a record for a read code indicating alcohol use, from which alcohol status could be derived‡ | Has a record of any drinking status recorded as read code, or captured through structured data areas in GP software§ | Has level of consumption recorded in read codes¶ | Has units per week recorded | Has had AUDIT** or FAST screen | Has AUDIT** or FAST score recorded | Has any level of drinking recorded in read, units or score†† |
| | N (col %)* | n (row %)† | n (row %)† | n (row %)† | n (row %)† | n (row %)† | n (row %)† | n (row %)† | n (row %)† |
| N | 1 768 651 (100) | 918 254 (51.9) | 897 535 (50.7) | 862 330 (48.8) | 558 747 (31.6) | 431 394 (24.4) | 204 182 (11.5) | 150 457 (8.5) | 862 642 (48.8) |
| **Sex** | | | | | | | | | |
| Male | 872 865 (49.4) | 416 611 (47.7) | 406 508 (46.6) | 389 946 (44.7) | 231 204 (26.5) | 222 678 (25.5) | 93 900 (10.8) | 69 661 (8.0) | 389 771 (44.7) |
| Female | 895 786 (50.6) | 501 643 (56.0) | 491 027 (54.8) | 472 384 (52.7) | 327 543 (36.6) | 208 716 (23.3) | 110 282 (12.3) | 80 796 (9.0) | 472 871 (52.8) |
| **Age group** | | | | | | | | | |
| 18–24 | 168 796 (9.5) | 58 379 (34.6) | 54 830 (32.5) | 51 584 (30.6) | 36 755 (21.8) | 16 537 (9.8) | 15 629 (9.3) | 8564 (5.1) | 51 647 (30.6) |
| 25–34 | 285 015 (16.1) | 131 905 (46.3) | 128 051 (44.9) | 119 772 (42.0) | 76 097 (26.7) | 50 686 (17.8) | 34 313 (12.0) | 25 222 (8.8) | 120 830 (42.4) |
| 35–44 | 300 819 (17.0) | 136 265 (45.3) | 133 289 (44.3) | 125 865 (41.8) | 77 277 (25.7) | 60 492 (20.1) | 34 273 (11.4) | 26 405 (8.8) | 126 761 (42.1) |
| 45–54 | 333 132 (18.8) | 171 021 (51.3) | 167 309 (50.2) | 160 742 (48.3) | 96 024 (28.8) | 88 593 (26.6) | 37 693 (11.3) | 28 709 (8.6) | 160 636 (48.2) |
| 55–64 | 277 217 (15.7) | 160 137 (57.8) | 157 180 (56.7) | 152 786 (55.1) | 94 656 (34.1) | 86 476 (31.2) | 32 478 (11.7) | 24 618 (8.9) | 151 987 (54.8) |
| 65–74 | 223 120 (12.6) | 145 105 (65.0) | 142 776 (64.0) | 139 320 (62.4) | 92 173 (41.3) | 77 896 (34.9) | 29 967 (13.4) | 22 483 (10.1) | 139 043 (62.3) |
| 75+ | 180 552 (10.2) | 115 442 (63.9) | 114 100 (63.2) | 112 261 (62.2) | 85 765 (47.5) | 50 714 (28.1) | 19 829 (11.0) | 14 456 (8.0) | 111 738 (61.9) |
| **Time since registration** | | | | | | | | | |
| 1–5 years | 334 629 (18.9) | 274 706 (82.1) | 266 784 (79.7) | 246 126 (73.6) | 150 578 (45.0) | 114 320 (34.2) | 102 332 (30.6) | 76 254 (22.8) | 253 079 (75.6) |
| >=5 years | 1 434 022 (81.1) | 643 548 (44.9) | 630 751 (44.0) | 616 204 (43.0) | 408 169 (28.5) | 317 074 (22.1) | 101 850 (7.1) | 74 203 (5.2) | 609 563 (42.5) |
| **Region** | | | | | | | | | |
| North England | 231 297 (13.1) | 138 506 (59.9) | 136 272 (58.9) | 128 522 (55.6) | 76 323 (33.0) | 76 396 (33.0) | 45 957 (19.9) | 40 689 (17.6) | 132 999 (57.5) |
| South England | 607 016 (34.3) | 320 297 (52.8) | 308 052 (50.7) | 285 002 (47.0) | 174 695 (28.8) | 143 027 (23.6) | 106 862 (17.6) | 77 866 (12.8) | 295 640 (48.7) |
| Northern Ireland | 114 311 (6.5) | 55 838 (48.8) | 53 203 (46.5) | 52 981 (46.3) | 34 830 (30.5) | 22 297 (19.5) | 13 363 (11.7) | 1104 (1.0) | 50 621 (44.3) |
| Scotland | 397 133 (22.5) | 207 247 (52.2) | 205 159 (51.7) | 201 171 (50.7) | 138 776 (34.9) | 103 816 (26.1) | 35 951 (9.1) | 30 202 (7.6) | 197 790 (49.8) |
| Wales | 418 894 (23.7) | 196 366 (46.9) | 194 849 (46.5) | 194 654 (46.5) | 134 123 (32.0) | 85 858 (20.5) | 2049 (0.5) | 596 (0.1) | 185 592 (44.3) |
| **Deprivation** | | | | | | | | | |
| 1 (least deprived) | 190 262 (10.8) | 102 692 (54.0) | 99 633 (52.4) | 90 969 (47.8) | 44 305 (23.3) | 57 838 (30.4) | 34 487 (18.1) | 27 071 (14.2) | 95 812 (50.4) |
| 2 | 134 191 (7.6) | 74 536 (55.5) | 72 437 (54.0) | 66 664 (49.7) | 35 727 (26.6) | 39 943 (29.8) | 25 722 (19.2) | 20 614 (15.4) | 69 844 (52.0) |
| 3 | 118 777 (6.7) | 67 455 (56.8) | 65 713 (55.3) | 61 953 (52.2) | 36 121 (30.4) | 33 892 (28.5) | 20 030 (16.9) | 15 365 (12.9) | 63 199 (53.2) |
| 4 | 108 991 (6.2) | 62 745 (57.6) | 61 689 (56.6) | 58 234 (53.4) | 39 674 (36.4) | 26 879 (24.7) | 19 040 (17.5) | 15 816 (14.5) | 59 403 (54.5) |
| 5 (most deprived) | 96 301 (5.4) | 57 840 (60.1) | 56 804 (59.0) | 51 770 (53.8) | 38 342 (39.8) | 21 983 (22.8) | 21 497 (22.3) | 18 212 (18.9) | 54 968 (57.1) |
| Missing | 1 120 129 (63.3) | 552 986 (49.4) | 541 259 (48.3) | 532 740 (47.6) | 364 578 (32.5) | 250 859 (22.4) | 83 406 (7.4) | 53 379 (4.8) | 519 416 (46.4) |

Continued

**Table 1** Continued

| | Study population: All individuals 18+ years registered with a CPRD practice on 1st January 2018* | Individuals in the study population with recording of alcohol use in different categories in the 5 years before 1st January 2018† | | | | | | | |
|---|---|---|---|---|---|---|---|---|---|
| | | Has a record for any read code relating to alcohol use | Has a record for a read code indicating alcohol use, from which alcohol status could be derived‡ | Has a record of any drinking status recorded as read code, or captured through structured data areas in GP software§ | Has level of consumption recorded in read codes¶ | Has units per week recorded | Has had AUDIT** or FAST screen | Has AUDIT** or FAST score recorded | Has any level of drinking recorded in read, units or score††† |
| | N (col %)* | n (row %)† | n (row %)† | n (row %)† | n (row %)† | n (row %)† | n (row %)† | n (row %)† | n (row %)† |
| **Ethnicity** | | | | | | | | | |
| White | 807 709 (45.7) | 502 125 (62.2) | 489 808 (60.6) | 465 333 (57.6) | 287 175 (35.6) | 254 409 (31.5) | 135 229 (16.7) | 104 643 (13.0) | 470 554 (58.3) |
| South Asian | 46 593 (2.6) | 28 089 (60.3) | 27 111 (58.2) | 25 603 (55.0) | 21 297 (45.7) | 5874 (12.6) | 9611 (20.6) | 6835 (14.7) | 26 306 (56.5) |
| Other | 60 520 (3.4) | 38 086 (62.9) | 36 482 (60.3) | 33 898 (56.0) | 26 559 (43.9) | 9974 (16.5) | 13 896 (23.0) | 10 120 (16.7) | 35 212 (58.2) |
| Missing | 853 829 (48.3) | 349 954 (41.0) | 344 134 (40.3) | 337 496 (39.5) | 223 716 (26.2) | 161 137 (18.9) | 45 446 (5.3) | 28 859 (3.4) | 330 570 (38.7) |
| **Smoking status** | | | | | | | | | |
| Non-smoker | 746 074 (42.2) | 383 302 (51.4) | 373 622 (50.1) | 356 424 (47.8) | 233 122 (31.2) | 163 733 (21.9) | 91 386 (12.2) | 66 152 (8.9) | 359 039 (48.1) |
| Current smoker | 254 850 (14.4) | 147 616 (57.9) | 144 275 (56.6) | 139 028 (54.6) | 90 505 (35.5) | 68 758 (27.0) | 31 617 (12.4) | 23 328 (9.2) | 137 108 (53.8) |
| Ex-smoker | 565 583 (32.0) | 347 945 (61.5) | 342 126 (60.5) | 332 036 (58.7) | 213 908 (37.8) | 181 109 (32.0) | 71 773 (12.7) | 53 832 (9.5) | 331 127 (58.5) |
| Missing | 202 144 (11.4) | 39 391 (19.5) | 37 512 (18.6) | 34 842 (17.2) | 21 212 (10.5) | 17 794 (8.8) | 9406 (4.7) | 7145 (3.5) | 35 368 (17.5) |
| **BMI** | | | | | | | | | |
| Normal (18.5–25 kg/m²) | 533 130 (30.1) | 298 415 (56.0) | 290 354 (54.5) | 277 070 (52.0) | 172 796 (32.4) | 139 540 (26.2) | 74 169 (13.9) | 54 609 (10.2) | 278 618 (52.3) |
| Underweight (BMI <18.5 kg/m²) | 32 421 (1.8) | 16 915 (52.2) | 16 262 (50.2) | 15 481 (47.7) | 10 904 (33.6) | 6055 (18.7) | 4327 (13.3) | 2938 (9.1) | 15 570 (48.0) |
| Overweight/obese (BMI ≥25 kg/m²) | 865 940 (49.0) | 543 118 (62.7) | 534 282 (61.7) | 519 336 (60.0) | 346 490 (40) | 265 441 (30.7) | 110 880 (12.8) | 82 856 (9.6) | 516 657 (59.7) |
| Missing | 337 160 (19.1) | 59 806 (17.7) | 56 637 (16.8) | 50 443 (15.0) | 28 557 (8.5) | 20 358 (6.0) | 14 806 (4.4) | 10 054 (3.0) | 51 797 (15.4) |
| Liver disease | 35 542 (2.0) | 25 404 (71.5) | 24 854 (69.9) | 24 340 (68.5) | 16 978 (47.8) | 12 847 (36.1) | 5012 (14.1) | 3684 (10.4) | 24 069 (67.7) |
| Hypertension | 358 285 (20.3) | 262 785 (73.3) | 260 050 (72.6) | 256 762 (71.7) | 183 616 (51.2) | 134 479 (37.5) | 44 646 (12.5) | 33 173 (9.3) | 254 136 (70.9) |
| Diabetes | 120 744 (6.8) | 100 787 (83.5) | 100 158 (83.0) | 99 378 (82.3) | 79 211 (65.6) | 47 668 (39.5) | 16 145 (13.4) | 12 164 (10.1) | 98 451 (81.5) |
| Depression | 118 644 (6.7) | 73 868 (62.3) | 72 314 (61.0) | 69 961 (59.0) | 47 499 (40.0) | 33 663 (28.4) | 15 581 (13.1) | 11 185 (9.4) | 69 083 (58.2) |
| Anxiety | 80 658 (4.6) | 50 048 (62.0) | 49 111 (60.9) | 47 593 (59.0) | 31 840 (39.5) | 23 330 (28.9) | 10 158 (12.6) | 7348 (9.1) | 47 006 (58.3) |

*For the column summarising characteristics for the whole study population (ie, all adults registered in CPRD on 1st January 2018) – percentages are out of the whole study population (ie, column %s).

†For subsequent columns — percentages are out of the whole study population (ie, column %s).

†For subsequent columns — which cover individuals identified through different types of alcohol-use recording — percentages represent the number of individuals recorded with the specific category of coding as a percentage of the study population in that particular stratum; that is, row percentage with denominator from study population in that stratum (eg, for all women in the study population (n=895 786), 56% (n=501 643) have a record of any Read code relating to alcohol use).

‡Alcohol status: current drinker, ex-drinker, non-drinker from Read codes.

§Alcohol status: Current drinker, ex-drinker, non-drinker from Read codes and captured through structured data areas in GP software.

¶Level of drinking: none, light, moderate, heavy drinker.

**Includes full AUDIT and short-form AUDIT-C screening.

†††Level of drinking derived from: Read codes, units per week or AUDIT/AUDIT-C/FAST screening score.

AUDIT, Alcohol Use Disorders Identification Test; BMI, body mass index; CPRD, Clinical Practice Research Datalink; GP, general practice.

**Table 2** Internal validation comparing measures of alcohol use in CPRD to audit and AUDIT-C scores, and to units per week. n=1 768 651

| | N (%) | N patients with AUDIT score on the same date as record N (%) | AUDIT score median | AUDIT score IQR | N patients with AUDIT-C score on the same date as record N (%) | AUDIT-C score median | AUDIT-C score IQR | N patients with units per week recorded on same date as record N (%) | Units per week median | Units per week IQR |
|---|---|---|---|---|---|---|---|---|---|---|
| **Current drinking status*** | | | | | | | | | | |
| Non | 174 970 (20.3) | 2571 (1.5) | 0 | 0–0 | 1022 (0.6) | 0 | 0–0 | 905 (0.5) | 0 | 0–1 |
| Ex | 31 526 (3.7) | 292 (0.9) | 0 | 0–0 | 39 (0.1) | 0 | 0–0 | 1004 (3.2) | 0 | 0–1 |
| Current | 655 834 (76.1) | 21 563 (3.3) | 3 | 1–5 | 4283 (0.7) | 3 | 2–5 | 333 374 (50.8) | 6 | 2–14 |
| **Current drinking level (Read codes)** | | | | | | | | | | |
| Non-drinker | 222 835 (39.9) | 2880 (1.3) | 0 | 0–0 | 1061 (0.5) | 0 | 0–0 | 2278 (1) | 0 | 0–2 |
| Light drinker | 255 919 (45.8) | 2962 (1.2) | 2 | 1–3 | 1058 (0.4) | 2 | 1–3 | 19 063 (7.4) | 3 | 1–8 |
| Moderate drinker | 52 070 (9.3) | 1122 (2.2) | 0 | 0–4 | 225 (0.4) | 3 | 2–4 | 12 753 (24.5) | 6 | 2–14 |
| Heavy drinker | 27 923 (5.0) | 2860 (10.2) | 6 | 5–9 | 125 (0.4) | 8 | 6–9 | 10 319 (36.9) | 18 | 9–30 |
| **Units per week** | | | | | | | | | | |
| 0 | 38 929 (9.0) | 1830 (4.7) | 0 | 0–0 | 123 (0.3) | 0 | 0–1 | n/a | n/a | |
| 1–14 | 298 441 (69.2) | 10 905 (3.7) | 3 | 2–4 | 1982 (0.7) | 3 | 2–4 | | | |
| 15–42 | 80 449 (18.6) | 2761 (3.4) | 6 | 4–8 | 364 (0.5) | 6 | 4–7 | | | |
| 43+ | 13 575 (3.1) | 222 (1.6) | 11 | 8–12 | 17 (0.1) | 8 | 6–10 | | | |
| **AUDIT-C score category** | | | | | | | | | | |
| 0–4; low risk | 16 206 (76.8) | 3994 (24.6) | 2 | 0–3 | n/a | n/a | | 1781 (11) | 4 | 2–8 |
| 5–12; high risk | 4893 (23.2) | 1129 (23.1) | 6 | 5–7 | | | | 702 (14.3) | 12 | 6–20 |
| **AUDIT score category (and corresponding WHO recommended intervention)[12]** | | | | | | | | | | |
| 0–7; alcohol education | 71 401 (92.5) | n/a | n/a | | 4881 (6.8) | 2 | 1–4 | 14 156 (19.8) | 4 | 2–10 |
| 8–15; simple advice | 5405 (7.0) | | | | 242 (4.5) | 9 | 8–10 | 1322 (24.9) | 20 | 14–30 |
| 16–19; simple advice plus brief counselling and continued monitoring | 206 (0.3) | | | | 0 (0) | – | – | 45 (21.8) | 35 | 20–50 |
| 20–40; referral to specialist for diagnostic evaluation and treatment | 200 (0.3) | | | | 0 (0) | – | – | 22 (11.2) | 55 | 32–70 |

*Current drinking status based on recorded status in Read codes and captured through structured data areas in GP software (not including derived status based on units consumed or AUDIT score).

AUDIT, Alcohol Use Disorders Identification Test; CPRD, Clinical Practice Research Datalink; GP, general practice; n/a, not available.

**Table 3** External validation, comparing most recent record of alcohol status in CPRD to data from the Health Survey for England 2016

| | CPRD | | HSE |
|---|---|---|---|
| | n | % | % |
| **Men** | | | |
| Non-drinker* | 76 544 | 20 | 17 |
| Current drinker | 313 402 | 80 | 83 |
| Units per week among current drinkers with units per week recorded: | | | |
| ≤14 (lower risk) | 122 787 | 67 | 63 |
| 15–50 (increased risk) | 53 923 | 30 | 31 |
| >50 (higher risk) | 5987 | 3 | 6 |
| **Women** | | | |
| Non-drinker* | 129 952 | 28 | 22 |
| Current drinker | 342 432 | 73 | 78 |
| Units per week among current drinkers with units per week recorded: | | | |
| ≤14 (lower risk) | 146 177 | 88 | 78 |
| 15–35 (increased risk) | 16 434 | 10 | 16 |
| >35 (higher risk) | 3094 | 2 | 5 |

*Includes patients labelled non- and ex-drinker.
CPRD, Clinical Practice Research Datalink; HSE, Health Survey for England.

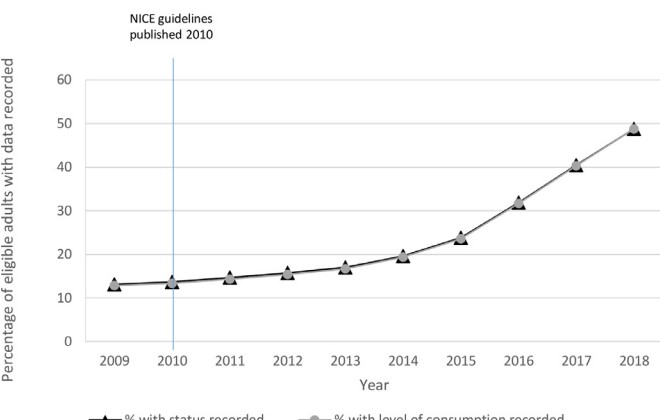

**Figure 1** Recording of alcohol status and time trends between 2009 and 2018. NICE, National Institute for Health and Care Excellence.

for a full AUDIT) the median AUDIT score was 6 (IQR 5 to 7).

### External validity

For men and women, the proportions of current and non-drinkers identified using Read-coding in CPRD were close to those estimated in Health Survey for England (HSE) (men: 80% current drinkers in CPRD, 83% in HSE; women: 73% current drinkers in CPRD, 78% in HSE). Units per week were also broadly comparable between CPRD and HSE although prevalence of higher risk drinking in CPRD looked slightly lower (table 3). However estimates for both the proportion of current drinkers and units consumed per week were slightly lower in CPRD than HSE.

### Time trends

Figure 1 illustrates time trends in recording of alcohol status and level of consumption. Between 2009 and 2018, there was an increase in recording in CPRD of both alcohol status and level of alcohol consumption from 19% to over 50%.

### Frequency of recording

Among patients with at least one alcohol record in the last 5 years, 48.9% had just one record, 21.8% had two records, 10.8% had three records and the remaining

48.5% had four or more records; 51.4% of patients with a record had their first alcohol record on the date of registration, 84.4% within the first month and 95.5% within the first year of registration.

For patients with more than one record of alcohol consumption, the median duration between records was 339 days (IQR 175 to 452). This was shorter for patients with heavy consumption (237 days (IQR 70 to 432) compared with light drinkers (median 351 (195 to 459 days)) or non-drinkers (median 335 (179 to 438 days)).

### Sensitivity analyses

1. Internal validity where we required that AUDIT score and units per week were recorded within 30 days of another code (rather than on the same date) resulted in more records being validated, but the results were identical to the main analysis (online supplementary appendix table 2).
2. Individuals with diabetes mellitus, hypertension and liver disease diagnosed in the 5 years prior to the study date had higher alcohol recording than shown in the main analysis (where these variables included all patients ever diagnosed) (diabetes 84.7% with recorded alcohol use in sensitivity analysis compared with 83.0% in main analysis with diabetes ever recorded; hypertension 75.8% with recorded alcohol use in sensitivity analysis, compared with 72.6% in main analysis and liver disease 77.1% with recorded alcohol use in sensitivity analysis, compared with 69.9% in main analysis).
3. Similarly, depression and anxiety diagnosed in the past year had higher levels of alcohol recording than when diagnosed in the previous 5 years (depression: 62.4% vs 61.0% in main analysis; anxiety: 62.0% vs 60.9% in main analysis).

### DISCUSSION
### Summary

In 2018, information about alcohol consumption was available for roughly half of all adult patients registered with practices contributing to the CPRD. Large differences

in recording were seen across patient characteristics, in particular increasing with age, deprivation and higher in patients with liver disease, hypertension, diabetes mellitus, depression and anxiety. Patients who were missing data on other risk factors such as smoking, BMI and ethnicity were also less likely to have any recording of alcohol use. Those with missing alcohol, smoking, BMI and ethnicity data may represent a group who have minimal contact with primary care. However, even among those with smoking recorded, general practitioners (GPs) only recorded drinking status in approximately 50% to 60% of patients. This suggests that GPs record alcohol use less frequently than smoking status; so even when there are opportunities to ask about health behaviours, alcohol use in not necessarily included. Alcohol was more likely to be recorded in those with higher levels of other health risks (smokers, obese) also suggesting that GPs may be more likely to ask about alcohol in patients who they perceive as at higher risk. These patients may also visit the GP more frequently, increasing the opportunities to ask about alcohol use.

The patterns of alcohol recording seen here show that UK GPs may be failing to record alcohol consumption at the most basic level of drinking status for half of their adult population. However, patients who are at greater risk from alcohol-related illness are being appropriately targeted (ie, those with mental health conditions and diabetes, liver disease, hypertension) and have better recording. Importantly, 20% to 40% of these at-risk patients still lack data on alcohol consumption and the proportion of patients screened for alcohol consumption appropriately using a valid screening tool is even lower (although one explanation for this observations might be that individuals scoring at low risk are screened but this information is not recorded). Alcohol use should be discussed as part of the cardiovascular disease health check for all patients over 40,[18] and suboptimal levels of alcohol recording are consistent with the low uptake of the health check recently described.[19] Importantly, just one-third of 18 to 24 year olds had alcohol consumption recorded, and they are particularly likely to engage in risky drinking behaviours.

Compared to a 2013 study using CPRD data (Khadjesari *et al*[7]), our study showed that while recording among new registrants has plateaued since 2009, remaining at around 75%, screening with AUDIT appears to have increased from 9% among new registrants in 2009 to 30.6% in our study. This may correspond with the introduction of new NICE guidance in 2010.[5] However, screening with a recommended screening tool for all registered adults remains at only 11%.

Our internal validation showed that, when there was a record of level of alcohol consumption, other measures of intake recorded in CPRD were likely to correspond. Recording of units per week corresponded to AUDIT scores (full AUDIT and AUDIT-C), and both measures corresponded to informal coding of alcohol consumption using Read codes (eg, codes for light/moderate/

heavy drinker) although the informal codes were less useful for discrimination between heavier drinkers (the median AUDIT score in those with a heavy drinker code was 6). AUDIT scores were used as the 'gold standard' because they are considered valid in comparison with other self-reported alcohol measures,[20 21] but all self-reported measures of alcohol use are limited as they are likely to be under-reported.[22]

The Health Survey for England found that, in 2016, 17% of men and 22% of women were teetotal. Using CPRD drinking status, 20.3% of patients were recorded as non-drinkers, so this is comparable and demonstrates external validity, although also self-reported. The internal and external validity demonstrated in this study give weight to any study using alcohol data from CPRD. While care must be taken to avoid introducing bias by restricting to only patients with alcohol recorded, the measures of consumption are likely to be useful in ranking participants in terms of their alcohol of consumption, with the caveats required by all self-reported alcohol research.

### Strengths and limitations

This study investigated alcohol recording across nearly 1.8 million UK adults, providing the power to explore recording in different subgroups of patients, and to understand in detail how alcohol use is recorded in primary care.

To validate, we used AUDIT scores recorded on the same date as the codes of interest. If an AUDIT questionnaire is completed, subsequent recording of alcohol consumption (in Read codes or units per week) might be more accurate than at a time when no scoring is done, and validity may be lower outside of these times. Additionally, GP practices that record both measures during the same consultation may be different to practices that only record one measure, and perhaps more likely to record alcohol use measures accurately. However, our sensitivity analyses using AUDIT scores recorded within 30 days before or after the date of the alcohol-related code of interest showed that records within a month of an AUDIT score or measure of units were internally consistent. For the external validation we used data from the Health Survey for England 2016, whereas the time frame for the prevalence estimates of alcohol use from CPRD were from 2013 to 2018 however although the time frames do not match perfectly 2016 is within the middle of the time frame and this seems an appropriate comparator.

Evidence suggests alcohol use in young people is common.[23] Indeed, the importance of screening and prevention in young people has been highlighted by recent NICE guidelines promoting alcohol intervention and education in the under 25s.[24] Unfortunately, we were unable to include 16 to 17 year olds in our analyses due to limited alcohol records in this age group and the resulting risk to anonymity. However, the finding of limited alcohol recording in 16 to 17 year olds is, in itself, an interesting insight into primary care alcohol recording in this vulnerable population.

Although we have demonstrated that alcohol recording and screening are higher for patients with certain high-risk conditions, it does not mean that GPs have acted on the screening result, or that they have screened at an appropriate time. For example, patients with depression or anxiety may not have been screened at the time when depression was diagnosed. We showed that the majority of screening took place within 1 month of registration, so whether this is repeated during consultations for high-risk conditions is not clear from these data.

Between 2009 and 2018, the proportion of CPRD patients with alcohol recorded more than tripled from around 13% to nearly 49%. It is likely that this increase in primary-care alcohol recording was contributed to by the 2010 NICE guidelines recommending screening on registration with a GP, and studies demonstrating the efficacy and cost-effectiveness of alcohol screening and brief interventions for alcohol misuse.[25] Screening is most likely to happen at the time of registration, shown here and in Khadjesari *et al*,[7] and therefore the increase in recording in our study may be linked to patient turnover. The result is that patients who have been at a practice for a long time are much less likely to have any record of their alcohol consumption; while recording has improved, there are clear and important differences between patients with and without a record for alcohol use.

### Implications for research

This study helps establish best practise in using primary care data to investigate alcohol use in research. Researchers using CPRD data should be aware of the limitations of alcohol data. Though the validity of the existing data has been described here, using only patients with complete alcohol use data may introduce selection bias into studies by preferentially including older, sicker and more deprived patients.

### Implications for practice

This study provides insight into how well aligned routine primary care practice is with NICE guidelines for alcohol screening. Recording of alcohol in primary care is not complete, meaning that opportunities for intervention may not be being identified. While we found evidence that alcohol use recording is higher in the high-risk groups highlighted by NICE (those with hypertension, depression), and at key times of patient registration recommended by NICE (among newly registered patients), there are still large numbers of patients in whom we might expect risky drinking who are not having their alcohol use recorded, and who therefore may not be targeted for intervention. In particular, younger individuals, in whom alcohol is a major cause of morbidity, maybe being overlooked.

**Contributors** EC and KM contributed equally. EH, SC, KM, LS, JKQ, EC and RD contributed to the design of the study. EH and EC extracted the data. EC, KM and EH wrote the statistical programmes and wrote the first draft. All authors (EH, SC, KM, EC, LS, JKQ, RD) contributed to further drafts and approved the final manuscript.

**Funding** EH holds a NIHR postdoctoral fellowship (grant number PDF-2016-09-029). LS received funding from the Wellcome Trust (Grant number 202912/B/16/Z).

**Disclaimer** The NIHR had no role in the design, analysis or writing up of this study.

**Competing interests** None declared.

**Patient consent for publication** Not required.

**Ethics approval** This study was approved by the London School of Hygiene and Tropical Medicine Ethics Committee (Approval Number 14454).

**Provenance and peer review** Not commissioned; externally peer reviewed.

**Data availability statement** Data may be obtained from a third party and are not publicly available.

**ORCID iDs**
Kathryn Mansfield http://orcid.org/0000-0002-2551-410X
Jennifer K Quint http://orcid.org/0000-0003-0149-4869

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
