## [Reviewer comments · BMJ Open]

ARTICLE DETAILS

TITLE (PROVISIONAL)	Completeness and validity of alcohol recording in general practice within the UK: a cross-sectional study
AUTHORS	Mansfield, Kathryn; Crellin, Elizabeth; Denholm, Rachel; Quint, Jennifer; Smeeth, Liam; Cook, Sarah; Herrett, Emily

VERSION 1 – REVIEW

REVIEWER	Karolina Kloda Independent Laboratory of Family Physician Education, Pomeranian Medical University in Szczecin, Poland
REVIEW RETURNED	25-May-2019

GENERAL COMMENTS	This study addresses the alcohol recording in primary care in UK. Although, authors presented the topic clearly, I have some major and minor comments. Major comments: 1. I am concerned that the AUDIT-C is not included in the analyzes. It's a recognized screening tool in primary care - general practice and several scientific projects, including ODHIN designed to improve the delivery of alcohol-related health care interventions, have used it. In my opinion, this is a significant limitation of this study and this should be discussed.2. The authors do not have enough evidence to be certain that the increase in alcohol recording is due to the NICE guidelines implementation. Such a relationship may exist, but other factors could also have an impact. For example, the previously mentioned ODHIN project (conducted in 2011-2014), as well as other projects aimed at increasing the activity of GPs in terms of alcohol screening and short intervention might be also important. Moreover, Figure 1 clearly presents that the notable increase of alcohol recording started not immediately after NICE guidelines publication, but from 2014.3. Limitations of the study should be presented more clearly, including lack of the AUDIT-C analysis. Minor comments: 4. Factors associated with alcohol recording are described twice - within Methods section and within Statistical analysis section. One of these descriptions should be omitted.5. Please provide the percentage of individuals of UK drinking at risky level within the Introduction section. Readers outside UK might be unfamiliar with UK population size.
---

REVIEWER	Andrew Thompson University of Liverpool, UK
REVIEW RETURNED	25-Jul-2019

GENERAL COMMENTS	This study takes a broad approach into assessing recorded information about alcohol consumption during primary care contact using CPRD. The data provides an overview of the coverage/granularity of alcohol consumption which has potential for informing future research design and highlighting areas/populations where basic health screening is being missed. Furthermore, the authors should be commended for their transparency (e.g. making available code lists and other additional information). There are however a number of areas where clarity/improvement is required: Introduction 1) P4 L8 – "...drinking at levels that risk their health" There is no level of drinking that doesn't risk carry some risk (e.g. low risk drinking). Methods 1) There is no clear rationale as to why the selected comorbidities were the only ones used. Although some are listed in NICE guidelines, this is by no means an exhaustive list. The rationales behind exclusions of epilepsy and previous stroke, for example, need to be carefully defined. The NICE guidelines also state 'other mood disorders' in relevant health problems – these disorders, however, have not been included in the present analysis. 2) It is interesting that the authors did not include persons aged 16/17 years in their analysis. This group is of specific interest – see Recommendation 7 of NICE guidelines. 3) Does point ii) in the outcomes include AUDIT-C? 4) Excluding AUDIT-C from the validation is an oversight. AUDIT-C can be used as a standalone or as a 'screening' tool to determine whether a full AUDIT assessment is required based on AUDIT-C score. This could have resulted in many low risk drinkers being excluded from the internal validation exercise. 5) P9 L10 – I believe this sentence needs to be reworked. Suggestion "We took the most recent alcohol records indicating alcohol use status (...) and/or units per work for each patient." Results 1) It would be interesting to know how many patients were registered for less than 5 years. 2) Can the authors provide the cross-over between patients with 'level of consumption' and 'drinking status' – there is only 0.1% between the absolute figures. 3) In addition to the proportion of patients with available data, it would also be useful to understand the mean/median time since last record. This would be very informative for individuals planning future research using this data. 4) P13 L41 – I am unsure where / how categorisation of AUDIT data occurred. Please clarify in methods, if not already available.
--

	5) P13 L54 – “... indicating need for referral...” 6) For those with multiple recordings (Freq of recording), what is the average time between these measures, and does this differ by level of consumption? Discussion 1) P16 L46 – the authors refer to patients as being at risk but fail to define to what this risk refers. 2) P16 L54 – “... although it is possible that individuals...” – is there any evidence to support this statement?
--	---

VERSION 1 – AUTHOR RESPONSE

Reviewer: 1

COMMENT 1.1

This study addresses the alcohol recording in primary care in UK. Although, authors presented the topic clearly, I have some major and minor comments.

Major comments:1. I am concerned that the AUDIT-C is not included in the analyzes. It's a recognized screening tool in primary care - general practice and several scientific projects, including ODHIN designed to improve the delivery of alcohol-related health care interventions, have used it. In my opinion, this is a significant limitation of this study and this should be discussed.

RESPONSE 1.1

We are grateful for your observation regarding the inclusion of AUDIT-C in our analyses. Our original results wrongly included AUDIT-C scoring with scores from the full AUDIT test. We have now separated out the AUDIT-C scores and included them as a separate measure of alcohol recording in CPRD.

MANUSCRIPT CHANGES 1.1

Location: Throughout manuscript (see particularly Methods p7, Results p13 & p14 and Tables 1 & 2)

COMMENT 1.2

2. The authors do not have enough evidence to be certain that the increase in alcohol recording is due to the NICE guidelines implementation. Such a relationship may exist, but other factors could also have an impact. For example, the previously mentioned ODHIN project (conducted in 2011-2014), as well as other projects aimed at increasing the activity of GPs in terms of alcohol screening and short intervention might be also important. Moreover, Figure 1 clearly presents that the notable increase of alcohol recording started not immediately after NICE guidelines publication, but from 2014.

RESPONSE 1.2

We agree, thank you for noting this, we have edited our discussion to clarify that the increase in primary care-based alcohol recording may have been driven by multiple factors, including the 2010 NICE guidelines and studies demonstrating the efficacy of alcohol screening and brief interventions for alcohol misuse.

MANUSCRIPT CHANGES 1.2

Location: Discussion (p19)

Between 2009 and 2018, the proportion of CPRD patients with alcohol recorded more than tripled from around 13% to nearly 49%. Factors that may have contributed to this increase in primary-care alcohol recording include the 2010 NICE guidelines recommending screening on registration with a GP, and studies demonstrating the efficacy and cost effectiveness of alcohol screening and brief interventions for alcohol misuse.

COMMENT 1.3

3. Limitations of the study should be presented more clearly, including lack of the AUDIT-C analysis.

RESPONSE 1.3

As in our response to Comment 1.1, we have addressed the AUDIT-C limitation by updating our analyses to include AUDIT-C scoring.

COMMENT 1.4

Minor comments:

4. Factors associated with alcohol recording are described twice - within Methods section and within Statistical analysis section. One of these descriptions should be omitted.

RESPONSE 1.4

We have edited the text in the Statistical analysis Section to reference the relevant text earlier in the Methods Section.

MANUSCRIPT CHANGES 1.4

Location: Methods (p8)

We then described alcohol recording prevalence in strata of individual characteristics (i.e. age, sex, and other factors described above under the heading 'Factors associated with alcohol recording').

COMMENT 1.5

5. Please provide the percentage of individuals of UK drinking at risky level within the Introduction section. Readers outside UK might be unfamiliar with UK population size.

RESPONSE 1.5

We have edited the Introduction appropriately.

MANUSCRIPT CHANGES 1.5

Location: Introduction (p4)

Hazardous alcohol use is a key behavioural risk factor affecting health. It is estimated that over 20% of adults the England are drinking at levels that are considered harmful,[1] with a resulting cost to the NHS of over £3 billion per year.[2]

Reviewer: 2

COMMENT 2.1

This study takes a broad approach into assessing recorded information about alcohol consumption during primary care contact using CPRD. The data provides an overview of the coverage/granularity of alcohol consumption which has potential for informing future research design and highlighting areas/populations where basic health screening is being missed. Furthermore, the authors should be commended for their transparency (e.g. making available code lists and other additional information). There are however a number of areas where clarity/improvement is required:

Introduction

1) P4 L8 – "...drinking at levels that risk their health" There is no level of drinking that doesn't risk carry some risk (e.g. low risk drinking).

RESPONSE 2.1

Thank you we have edited appropriately to avoid any misunderstanding.

MANUSCRIPT CHANGES 2.1

Location: Introduction (p4)

Hazardous alcohol use is a key behavioural risk factor affecting health. It is estimated that over 20% of adults the England are drinking at levels that are considered harmful,[1] with a resulting cost to the NHS of over £3 billion per year.[2]

COMMENT 2.2

Methods

1) There is no clear rationale as to why the selected comorbidities were the only ones used. Although some are listed in NICE guidelines, this is by no means an exhaustive list. The rationales behind exclusions of epilepsy and previous stroke, for example, need to be carefully defined. The NICE guidelines also state 'other mood disorders' in relevant health problems – these disorders, however, have not been included in the present analysis.

RESPONSE 2.3

Thank you, we selected only a few relevant comorbidities that we felt we could reliably capture using electronic health record data from primary care. You are right in stating that this is not an exhaustive list, we have therefore explicitly edited our Methods section to clarify that we are only considering a selected list of comorbidities.

MANUSCRIPT CHANGES 2.1

Location: Methods (p6)

We investigated a range of potential factors that may be associated with alcohol-use recording in primary care: age, sex, deprivation, geographic region, selected comorbidities related to alcohol use that could be captured in electronic health record data from a primary care setting (liver disease, hypertension, depression and anxiety) or those where health function monitoring is likely to be more frequent/complete (diabetes mellitus), time since registration at the current practice (1-5 years vs ≥5 years), body mass index (BMI), smoking status, and ethnicity.

COMMENT 2.3

2) It is interesting that the authors did not include persons aged 16/17 years in their analysis. This group is of specific interest – see Recommendation 7 of NICE guidelines.

RESPONSE 2.3

We agree, that it would be interesting to consider this age group, indeed we did initially include people 16-17 in our analyses. However, there were only a limited number of records for screening within this age group and presenting their data risked compromising anonymity and would be against the guidelines of the data provider to publish. We have added some text in our discussion presenting this as a limitation, as we recognise that this in itself is an important finding.

MANUSCRIPT CHANGES 2.3

Location: Discussion (p19)

Evidence suggests alcohol use in young people is common.[23] Indeed, the importance of screening and prevention in young people has been highlighted by recent NICE guidelines promoting alcohol intervention and education in the under 25s.[24] Unfortunately, we were unable to include 16-17 year olds in our analyses due to limited alcohol records in this age group and the resulting risk to anonymity. However, the finding of limited alcohol recording in 16-17 year olds is, in itself, an interesting insight into primary care alcohol recording in this vulnerable population.

COMMENT 2.4

3) Does point ii) in the outcomes include AUDIT-C?

RESPONSE 2.4

Yes, Outcome Number ii (Codes indicating AUDIT or FAST screening) in our alcohol-recording outcomes does include individuals with a record for AUDIT-C (in addition to records indicating the full AUDIT or FAST screening). We have edited the Outcomes text to clarify this.

MANUSCRIPT CHANGES 2.4

Location: Methods (p7)

ii. Codes indicating AUDIT (including short-form AUDIT-C and full AUDIT) or FAST screening.

COMMENT 2.5

4) Excluding AUDIT-C from the validation is an oversight. AUDIT-C can be used as a standalone or as a 'screening' tool to determine whether a full AUDIT assessment is required based on AUDIT-C score. This could have resulted in many low risk drinkers being excluded from the internal validation exercise.

RESPONSE 2.5

We agree. Please see our response to Reviewer 1, Comment 1.1.

COMMENT 2.6

5) P9 L10 – I believe this sentence needs to be reworked. Suggestion “We took the most recent alcohol records indicating alcohol use status (...) and/or units per week for each patient.”

RESPONSE 2.6

Thank you, we have edited the relevant section for clarity.

MANUSCRIPT CHANGES 2.6

Location: Methods (p9)

We identified current alcohol-use status for each individual with a relevant primary care alcohol record by identifying the most recent record indicating alcohol intake; i.e. either a morbidity code indicating alcohol-use status (classified as current or non-/ex-drinkers) or a record indicating the number of units of alcohol consumed per week.

COMMENT 2.7

Results

1) It would be interesting to know how many patients were registered for less than 5 years.

RESPONSE 2.7

There were 334,629 participants registered for less than 5 years (18.9% of the sample). This is shown in Table 1.

COMMENT 2.8

2) Can the authors provide the cross-over between patients with 'level of consumption' and 'drinking status' – there is only 0.1% between the absolute figures.

RESPONSE 2.8

We have broken down recording of drinking status in Table 1 into two categories: 1) "Has a record for a READ code from which drinking status could be derived"; and 2) "Has a record of any drinking status recorded as Read code, or captured through structured data areas in GP software". The second category is a subset of the first category, which is why there is a very high overlap. We have modified Table 1 to make this clearer.

MANUSCRIPT CHANGES 2.8

Location: Table 1

Changed column heading to, "Has a Read code indicating alcohol use, from which drinking status could be derived".

COMMENT 2.9

3) In addition to the proportion of patients with available data, it would also be useful to understand the mean/median time since last record. This would be very informative for individuals planning future research using this data.

RESPONSE 2.9

We have now included this in the results section

MANUSCRIPT CHANGES 2.9

Location: Results (p15)

For patients with more than one record of alcohol consumption, the median duration between records was 339 days (IQR 175-452). This was shorter for patients with heavy consumption (237 days (IQR 70-432) compared with light drinkers (median 351 (195-459 days)) or non-drinkers (median 335 (179-438 days)).

COMMENT 2.10

4) P13 L41 – I am unsure where / how categorisation of AUDIT data occurred. Please clarify in methods, if not already available.

RESPONSE 2.10

We categorised AUDIT score, according to WHO guidelines, as: non-drinker or low- risk drinking (0-7); hazardous drinking (8-15); harmful drinking (16-19); and possible dependence (20-40). This is presented in the Methods Section (p7). We have also included some additional detail where we report the results of analyses by AUDIT score category (both in the manuscript and Table 2).

MANUSCRIPT CHANGES 2.10

Location: Results (p13)

AUDIT scores: the majority (92.5%) of patients had scores in the lowest category (AUDIT score 0-7, where alcohol education is recommended by WHO guidelines[11]). Patients with higher AUDIT scores consumed more units per week; for example, those in the lowest AUDIT score category (0-7,

alcohol education) had a median units per week of four (IQR 2-10) compared to a median of 55 units per week (IQR 32-70) in the highest AUDIT score category (score 20-40, indicating need for referral to specialist for diagnostic evaluation and treatment according to WHO guidelines[11]).

Location: Table 2, edit to row header to reference WHO intervention guidelines for WHO categories of AUDIT score.

COMMENT 2.11

5) P13 L54 – "... indicating need for referral..."

RESPONSE 2.11

Please see our response to your previous comment (Comment 2.10).

COMMENT 2.12

6) For those with multiple recordings (Freq of recording), what is the average time between these measures, and does this differ by level of consumption?

RESPONSE 2.12

We have added this to the results section

MANUSCRIPT CHANGES 2.9

Location: Results (p15)

For patients with more than one record of alcohol consumption, the median duration between records was 339 days (IQR 175-452). This was shorter for patients with heavy consumption (237 days (IQR 70-432) compared with light drinkers (median 351 (195-459 days)) or non-drinkers (median 335 (179-438 days)).

COMMENT 2.13

Discussion

1) P16 L46 – the authors refer to patients as being at risk but fail to define to what this risk refers.

RESPONSE 2.13

We have edited the text appropriately.

MANUSCRIPT CHANGES 2.13

Location: Discussion (p16)

However, patients who are at greater risk from alcohol-related illness are being appropriately targeted (i.e. those with mental health conditions and diabetes, liver disease, hypertension) and have better recording

COMMENT 2.14

2) P16 L54 – "... although it is possible that individuals..." – is there any evidence to support this statement?

RESPONSE 2.14

No, we have no evidence for this statement, this is simply one explanation for what we observed. We have edited the text to clarify that this.

MANUSCRIPT CHANGES 2.14

Location: Discussion (p16)

Importantly, 20-40% of these at-risk patients still lack data on alcohol consumption and the proportion of patients screened for alcohol consumption appropriately using a valid screening tool is even lower (although one explanation for this observation might be that individuals scoring at low risk are screened but this information is not recorded).

VERSION 2 – REVIEW

REVIEWER	Karolina Kloda Pomeranian Medical University in Szczecin, Poland
REVIEW RETURNED	25-Sep-2019

GENERAL COMMENTS	Thank you for addressing all my concerns. This manuscript is a valuable source of information in its present form.
--

REVIEWER	Andrew Thompson Senior Research Associate / MRC Fellow University of Liverpool UK
REVIEW RETURNED	19-Sep-2019

GENERAL COMMENTS	The authors have made changes to address the concerns/recommendations of both reviewers. I have a few additional minor points that might help improve the final version of the manuscript: 1) Although referenced earlier, I would add the reference for the NICE PH24 guidelines to the final paragraph of the Introduction. 2) In the presentation of the sensitivity analyses, the use of language is not consistent regarding changes between the sensitivity analysis outcomes and those from the main analysis. In 2. 'slightly higher' is used and in 3. 'higher', when the absolute differences are similar (e.g. change in diabetes is 1.7% and depression is 1.4%).
--

VERSION 2 – AUTHOR RESPONSE

Reviewer: 1

COMMENT 1.1

Thank you for addressing all my concerns. This manuscript is a valuable source of information in its present form.

RESPONSE 2.1

Thank you for your time and comments.

Reviewer: 2

COMMENT 2.1

The authors have made changes to address the concerns/recommendations of both reviewers. I have a few additional minor points that might help improve the final version of the manuscript:

Although referenced earlier, I would add the reference for the NICE PH24 guidelines to the final paragraph of the Introduction.

RESPONSE 2.1

We have now added this reference in the final paragraph of the Introduction.

COMMENT 2.2

In the presentation of the sensitivity analyses, the use of language is not consistent regarding changes between the sensitivity analysis outcomes and those from the main analysis. In 2. 'slightly higher' is used and in 3. 'higher', when the absolute differences are similar (e.g. change in diabetes is 1.7% and depression is 1.4%).

RESPONSE 1.2

We have now amended the text in the Results section; we have removed the word 'slightly' to be more consistent.